



# A new Greenland digital elevation model derived from ICESat-2

Yubin Fan[1,2,3], Chang-Qing Ke[1,2,3*], Xiaoyi Shen[1,2,3]

[1]Jiangsu Provincial Key Laboratory of Geographic Information Science and Technology, Key Laboratory for Land Satellite Remote Sensing Applications of Ministry of Natural Resources, School of Geography and Ocean Science, Nanjing University, Nanjing, 210023 China.
[2]Collaborative Innovation Center of Novel Software Technology and Industrialization, Nanjing, 210023 China.
[3]Collaborative Innovation Center of South China Sea Studies, Nanjing, 210023 China.

*Correspondence to*: Chang-Qing Ke (kecq@nju.edu.cn)

**Abstract.** Greenland digital elevation models (DEMs) are indispensable to fieldwork, ice velocity calculations and mass change estimations. Previous DEMs provided Greenland elevation information for different periods, but long temporal coverage introduced additional time uncertainty to scientific research. To provide a high-resolution DEM with a definite time, approximately $5.8 \times 10^8$ ICESat-2 observations from November 2018 to November 2019 were used to generate a new DEM for both the ice sheet and glaciers in peripheral Greenland. A spatiotemporal model fit process was first performed at 500 m resolution. To improve ICESat-2 data utilization, DEMs with 1 km and 2 km resolution across all of Greenland and an additional 5 km resolution in southernmost Greenland were used to fill the DEM gaps. Kriging interpolation was used to fill the remaining 2% of void grids that were insufficiently observed by ICESat-2 measurements. IceBridge mission data acquired by the Airborne Topographic Mapper (ATM) Lidar system were used to evaluate the accuracy of the newly generated ICESat-2 DEM. Overall, the ICESat-2 DEM had a median difference of -0.48 m for all of Greenland, which agreed well with the IceBridge data, and the performance in the calculated and interpolated grids was similar. Better accuracy could be observed in the northern basins due to the larger proportion of calculated grids with 500 m resolution. The ICESat-2 DEM showed significant improvements in accuracy compared with other altimeter-derived DEMs. Compared to the DEM generated by image pairs, the accuracy was also significantly higher than those of the 1 km ArcticDEM and TanDEM. Similar performance between the ICESat-2 DEM and 500 m ArcticDEM indicated the high accuracy and reliability of the ICESat-2 DEM. Moreover, the ICESat-2 DEM performed better on northern aspects than the 500 m ArcticDEM. Overall, the ICESat-2 DEM showed great accuracy stability under various topographic conditions, hence providing a time-accurate DEM with high accuracy that will be helpful to study elevation and mass balance changes in Greenland. The Greenland DEM and its uncertainty are available at (https://data.tpdc.ac.cn/en/disallow/07497631-0475-48b5-ba53-c17f9076c72f/, Fan et al, 2021).

## 1 Introduction

The digital elevation model (DEM) of Greenland is particularly important for fieldwork planning, numerical modelling verification, and ice movement tracking (Bamber et al., 2009; Bamber et al., 2013). Combining the DEM with the ice thickness data, the ice deformation rate and the underlying bedrock condition can be measured, which can be used to





determine subglacial hydrological pathways (Bamber et al., 2013). The surface elevation at different periods is also indispensable for studying elevation and mass changes to understand ice dynamics and estimate potential sea level changes

(Sutterley et al., 2014; Smith et al., 2020). In addition, InSAR estimation of ice velocity requires high-accuracy and up-to-date DEMs to distinguish phase differences caused by terrain and ice sheet movement (Riel et al., 2021).

The previously published Greenland DEM dates back to the 1980s, and 3500 photographs obtained from 1978 to 1987 were used to provide the elevations of peripheral Greenland with a resolution of 25 m (Korsgaard et al., 2016). However, the data acquisition was limited by the low-visibility contrast between snow and ice surfaces (Noh and Howat, 2015), which

introduced large time uncertainty into the DEM. Research regions were also restricted to the margin and outlets of Greenland, and there is a lack of understanding about the internal Greenland ice sheet.

More DEM products became available to the public in the past 20 years with the improvement in observation methodologies. ASTER and SPOT 5 measurements were used on the ice sheet periphery, and AVHRR photoclinometry was used in the inner and polar regions to derive the Greenland Ice Mapping Project (GIMP) DEM (version 1) (Howat et al., 2014). The

GIMP1 DEM was then vertically calibrated by the Ice, Cloud, and Land Elevation Satellite (ICESat) and had a mean difference of -0.72 m with respect to the contemporaneous IceBridge data (Xing et al., 2020). ICESat had the advantages of wide coverage (86°N-86°S), high single-point accuracy (0.1-0.15 m), and small footprint size (70 m) compared with the former radar altimeter (e.g., Envisat, footprint size: 2–10 km, coverage: 81.5°N-81.5°S) (Zwally et al., 2002), enhancing the ability to measure the elevation of Greenland. Hence, the ICESat DEM adopted a bi-quadratic surface to fit all ICESat

footprints within each 1 km grid, but the largest radius of 20 km in the low-latitude regions to some extent limited the ability to describe the small-scale elevation patterns at the Greenland margin (DiMarzio et al. 2007).

Benefitting from the ability to penetrate clouds, radar data can provide elevation information unaffected by weather conditions. CryoSat-2 carried a Ku-band synthetic aperture interferometric radar altimeter (SIRAL), further increasing the spatial coverage to 88°N-88°S. Although the footprint size (approximately 300 m) was larger than that of ICESat, the

smaller cross-track distance (2.5 km) significantly improved its ability to monitor the ice sheet compared with that of ICESat (25 km) (Wingham et al., 2002). Thanks to the above advantages, CryoSat-2 L1B level data from 2011 to 2014 provided a reliable data source to provide elevation estimates (Helm et al., 2014). TanDEM-X and TerraSAR-X, high-resolution and all-weather SAR data, were used to generate the Greenland DEM using differential interferometry (Zink et al., 2014), but the radar signal can penetrate into the snow, which causes the elevation to be underestimated.

The GIMP2 elevation dataset was generated by the high-resolution panchromatic stereoscopic images of the GeoEye-1, WorldView-1, WorldView-2, and WorldView-3 satellites, and the time scale of the DEM was from 2009 to 2015. ArcticDEM, the latest released DEM, which was generated by the same sources as GIMP2, has the highest resolution (2 m) among all free available polar DEMs. However, it is difficult to screen optical image pairs as DEM data sources because of the influence of weather, clouds, and the solar elevation angle (Korona et al., 2009). As a result, the mosaicked DEM is the

combination of images from many times, and it is difficult to quantify the exact time of the DEM, limiting its scientific applications.



ICESat-2, a new generation of satellite-borne Lidar altimeters, is intended as a successor to the ICESat mission to quantify the contribution of polar ice sheets to sea level rise and the impact of climate change (Markus et al., 2017). ICESat-2 has an orbital altitude of 500 km and an orbital inclination of 92°, accompanied by a revisit period of 91 days, which provides

centimetre-scale measurements of different surface types. The ICESat-2 beam footprint is approximately 17 m, with a spatial interval of 0.7 m (Neumann et al., 2019), which ensures accurate measurements of elevation at a high orbital resolution by determining the local ice sheet slope. A much finer observation can be obtained owing to its along-track distance of 0.7 m and cross-track distance of 3.3 km, which is a great improvement over CryoSat-2's along-track distance of 1.5 km and cross-track distance of 3 km, and ICESat's along-track distance of 170 m. Not only the resolution but also the accuracy is improved.

The accuracy in the flat ice sheet can reach 3 cm, and the accuracy can still be less than 14 cm even for complex topography (Shen et al., 2021), which makes ICESat-2 a great data source to generate a DEM with high resolution and accuracy.

Hence, we present a newly generated Greenland DEM with a resolution of 500 m using a spatiotemporal model fit, which is based on ICESat-2 measurements from November 2018 to November 2019. IceBridge data were used to evaluate the accuracy for all of Greenland and for different basins. The performance was also compared with other published DEMs

under various terrain conditions to validate the reliability of the DEM.

## 2 Data

### 2.1 ICESat-2 ATL06 data

The ICESat-2 land ice height product ATL06 (Release 003) was used here for DEM generation. The product provides longitude, latitude, and surface heights based on the WGS84 ellipsoid and data acquisition time. The ATL06 product is

developed from global geo-located photon data (ATL03) to estimate the land ice height (Smith et al., 2019). Compared with the original ATL03 product, land ice height is determined after instrument bias corrections (e.g., transmit pulse shape bias correction and first-photon bias correction) (Markus et al., 2017). The beam pair separation of the ATL06 product is set at 3.3 km across the track. The three pairs contain one strong beam and one weak beam, and the two beams within each pair are separated by 90 m.

The signal of the strong beams is stronger than that of the weak beams, so strong beams can provide more accurate measurements than weak beams. However, for strong and weak beams in the ATL06 product, both beams in one pair show similar performance, with a median difference of -0.08 cm and -0.13 cm for strong beam2 and weak beam1 (Shen et al., 2021). Although weak beams do not perform as well as strong beams, the accuracy is still higher than that of other altimeters. CryoSat-2, for example, has an accuracy of approximately 50 cm over the ice shelves and the interior of the ice sheet, with

an error of more than 4 m on slopes greater than 0.9° (Wang et al., 2015). Hence, we did not exclude weak beams considering increased spatial coverage and data point utilization since no systemic error was found in ICESat-2 elevation measurements by different beams. However, only data marked as good quality (atl06_quality_summary=0) were used for





DEM generation to improve the accuracy of the DEM. Over the entire Greenland ice sheet and ice caps, we used approximately $5.8 \times 10^8$ ICESat-2 elevation footprints to generate a new DEM, that is, the ICESat-2 DEM..

## 2.2 IceBridge data

To evaluate the accuracy of the ICESat-2 DEM, ATM surface elevation data from the IceBridge survey were used. ATM was intended to fill the gap between ICESat and ICESat-2, working at the same wavelength (532 nm) as ICESat-2. The absolute elevation accuracy of the ATM system can reach 0.1 m, and the position accuracy on the flat ice sheet is less than 1 m (Kurtz et al., 2013). The IceBridge ATM L2 Icessn Elevation, Slope, and Roughness Version 2 dataset was used to evaluate the DEMs. The final resolution of IceBridge was resampled to 25 m, and the estimated error was approximately 12 cm (Krabill et al., 2004). The root-mean-square error (RMSE) was taken as the roughness of each IceBridge data. The slope and aspect were calculated as follows (Shen et al., 2021).

$$\alpha = \arctan(\alpha_{s,n}^2 + \alpha_{w,e}^2) \times \frac{180}{\pi} \tag{1}$$

$$\beta_0 = \arctan 2(\alpha_{s,n}, \alpha_{w,e}) \times \frac{180}{\pi} \tag{2}$$

$$\arctan 2(y, x) = \begin{cases} \arctan\frac{y}{x}, & x > 0 \\ \arctan\frac{y}{x} + \pi, & y \geq 0, x < 0 \\ \arctan\frac{y}{x} - \pi, & y < 0, x < 0 \\ \frac{\pi}{2}, & y > 0, x = 0 \\ -\frac{\pi}{2}, & y < 0, x = 0 \end{cases} \tag{3}$$

$$\beta = \begin{cases} \frac{\pi}{2} - \beta_0, \beta_0 < \frac{\pi}{2} \\ \frac{3\pi}{2} - \beta_0, \beta_0 > \frac{\pi}{2} \end{cases} \tag{4}$$

During 2009-2019, IceBridge provided millions of footprints over Greenland, which cover both the peripheral and inland areas of Greenland. The distribution of IceBridge data for May 2019, which was used to evaluate the accuracy of the new ICESat-2 DEM, is displayed in Figure 1. We also calculated the histogram of the elevation, surface slope, surface aspect, and roughness of IceBridge in May 2019. Overall, the elevations of the sampled regions ranged from 0 m to 3500 m, the surface slopes ranged from 0° to 10°, the surface aspects ranged from 0° to 360°, and the roughnesses ranged from 0 cm to 20 cm (**Figure 2**). These sampled areas had variable surface terrain conditions, which provided a reliable dataset to evaluate the performance of the ICESat-2 DEM.

## 2.3 Currently available other Greenland DEMs

### 2.3.1 GLAS/ICESat 1 km Laser Altimetry DEM

Greenland's DEM, derived from GLAS/ICESat laser altimetry data (from February 2003 to June 2005), provides the surface elevation for both Greenland ice sheets and caps, with less impact on slopes compared with radar altimetry data such as



EnviSat and ERS 1/2. The spatial resolution is 1 km. The horizontal coordinates are based on polar stereographic coordinates, and elevations are provided with respect to the WGS84 ellipsoid.

### 2.3.2 ArcticDEM

ArcticDEM is a high-resolution, high-quality digital surface model (DSM) of the Arctic, with the highest resolution of 2 m, followed by resolutions of 10 m, 32 m, 100 m, 500 m, and 1000 m. The temporal coverage of the ArcticDEM project is mainly from 2015 to 2018. The mosaicked DEM files are compiled from the best quality strip DEM files, and the filtered ICESat altimetry data are applied to improve the absolute accuracy. The estimated accuracy is approximately 85 cm at a resolution of 100 m (Xing et al., 2020). We used the elevation products of 500 m and 1000 m for comparisons.

### 2.3.3 TanDEM DEM

The TanDEM-X DEM is a global DEM with a resolution of 90 m provided by the German Aerospace Centre (DLR). Data collection was completed in 2015, and global DEM production was completed in 2016 and published in 2018. Different from previous datasets, it was generated by two X-band radar satellites (TanDEM-X and TerraSAR-X), which provide synchronous information to create accurate elevation information about the Earth's land surface. The absolute horizontal and vertical accuracy can reach less than 10 m. The temporal coverage of the TanDEM data is mainly from 2011 to 2014.

### 2.3.4 CryoSat-2 DEM

CryoSat-2 L1B level data from January 2011 to January 2014 were used to provide the elevation of Greenland. Based on CryoSat-2 L1B level data from 2011 to 2014, waveform re-tracking and interferometry were performed on LRM and SARIn data, respectively, and slope correction was applied to the original product to improve the elevation estimates (Helm et al., 2014). The accuracy of the CryoSat-2 DEM was less than 1 m in flat regions and less than 4 m in rugged regions, showing similar performance to the other DEMs obtained by laser and radar altimeters.

## 3 Methods

### 3.1 DEM generation

To compute the elevation of Greenland, we followed the method of Slater et al. (2018), which is a spatiotemporal model to fit the surface height in each grid. The model used is described as follows:

$$\mathrm{h}_i = h + a_0 x + a_1 y + a_2 x^2 + a_3 y^2 + a_4 xy + \frac{\mathrm{dh}}{\mathrm{dt}}(t - t_{mid})$$

$$(5)$$



where hi represents the modelled elevation, dh/dt is the elevation change rate in the 13 months, t is the ith month starting from November 2018, tmid is the time of the mid timestamp (May 2019), and (x,y) are the coordinates in the polar
stereographic projection. The uncertainty is the 95% confidence level for elevation.

The selection criterion for DEM resolution is to depict the detailed elevation pattern, so finer resolution should be adopted. A much finer resolution can reveal more detailed elevation patterns; this will lead to more calculated gaps, making the proportion of interpolated grids larger and leading to larger biases and uncertainties. The proportions of the calculated grids are latitude-dependent, and high-latitude areas account for higher coverage of the calculated grids, owing to their dense
orbital distribution. However, it is difficult to increase the proportion of calculated grids in low-latitude areas by using high spatial resolution.

Hence, we tested resolutions of 250 m, 500 m, 1 km, 2 km, and 5 km to achieve the optimal resolution. We found that the 250 m grid covered only 15.38% of the Greenland area and only 30% even at high latitudes, so we discarded this resolution for further processing. In contrast, a 500 m resolution increases overall coverage to 33% and coverage at high latitudes to
nearly 70% (Figure 3). With a 1 km resolution, the proportion of calculated grids exceeds 90% in the basins that are north of 75°N (basins 1, 2, and 8). The resolution of 2 km covers 82% of all Greenland. However, a 2 km resolution cannot obtain optimal coverage in low-elevation areas, while a 5 km resolution can further increase the coverage of calculated grids to 98%, especially in the southern basins (basins 4, 5, and 6).

Ultimately, we separated Greenland into $8.5 \times 10^6$ grids to compute the elevation in a 500 m×500 m grid. To build a fine-
resolution DEM and decrease the interpolation coverage as much as possible, the 1 km and 2 km results were used to fill the observed gaps in the DEM by resampling their results to 500 m by bilinear interpolation. The same process was repeated at a 5 km resolution in southernmost Greenland. The results of each resolution were constrained in terms of data availability, data quality, and rationality. We set the minimum number of grid points to 10 and the minimum timestamp to 2 months, which could ensure that enough measurements were contained in a grid cell to make a good elevation estimation. In addition, we
introduced thresholds to remove outliers, which are RMSE≥10 m, the uncertainty of elevation change ≥10 m, the rate of elevation change≥10 m/yr and the rate of elevation change uncertainty ≥0.4 m/yr (Slater et al., 2018). The original kriging method was adopted to fill the 2% data holes that still existed. To reduce the influence of different resolutions, a median filter of 2.5 km×2.5 km was applied to the final ICESat-2 DEM.

**3.2 DEM accuracy evaluation**

First, we grouped IceBridge data according to different DEM grids and calculated the median elevation, slope, aspect, and roughness within each grid, taken as the true values in this grid. The height differences were obtained by subtracting the DEM elevation at the location of IceBridge from the median IceBridge elevation. Subsequently, we used the median difference (MED), the mean difference (MD), the median absolute difference (MAD), the standard deviation (STD), the




RMSE, the 90% confidence interval LE$_{90}$ *(Gonzalez-Moradas and Viveen, 2020; Höhle and Höhle, 2009)*, and the

correlation (R) to evaluate each DEM. The calculations are as follows:

$$dh = \text{median(IceBridge)} - \text{DEM} \tag{6}$$

$$\text{MED} = \text{median(dh)} \tag{7}$$

$$\text{MD} = \frac{1}{n}\sum_{i=1}^{n} dh_i \tag{8}$$

$$\text{MAD} = \text{median(|dh|)} \tag{9}$$

$$\text{STD} = \sqrt{\frac{\sum_{i=1}^{n}(dh_i - \text{MD})}{n-1}} \tag{10}$$

$$\text{RMSE} = \sqrt{\frac{\sum_{i=1}^{n} dh_i^2}{n-1}} \tag{11}$$

$$\text{LE}_{90} = 1.6449 \times \text{STD} \tag{12}$$

where $dh_i$ is the elevation difference in each DEM grid and *n* is the number of overlapping IceBridge footprints.

We additionally used the elevation intervals of 0 m to 500 m, 500 m to 1000 m, 1000 m to 1500 m, 1500 m to 2000 m, and

≥2000 m to study the relationship between the elevation difference and elevation. For the surface slope, we divided the

slope into 5 intervals of 0° to 0.25°, 0.25° to 0.5°, 0.5° to 1°, 1° to 2°, and ≥2° to detect the relationship between the

elevation difference and slope. Similarly, the same step was repeated for roughness intervals of 0 cm to 5 cm, 5 cm to 10 cm,

10 cm to 15 cm, 15 cm to 20 cm, and ≥20 cm. We identified the aspect as north, east, south, and west to investigate the

relationship between the elevation difference and terrain aspect.

195 **4. Results**

**4.1 General attributes of ICESat-2 DEM**

Approximately 33.00%, 23.93%, and 25.43% of elevations were directly estimated from ICESat-2 at 500 m, 1 km, and 2 km

resolutions, corresponding to the ICESat-2 measurements of $3.51\times10^8$, $3.96\times10^8$, and $4.50\times10^8$, respectively. The remaining

data could not be covered in DEM grids because the observations were insufficient. Although ICESat-2 has denser coverage

200 than previous satellite altimeters, not all grids were estimated from the ICESat-2 measurements. Hence, the other grids were

interpolated from the neighbouring elevations by using the ordinary kriging method.

The ICESat-2 DEM shows the same pattern as the other published Greenland DEMs, with the highest elevation appearing in

the interior ice sheets and showing a downward trend to the margins. Lower elevations occur on the surrounding individual

glaciers around the periphery of Greenland, which also exhibits large topographic fluctuations. In addition to the DEM, the

205 model fitting method can also be used to obtain the monthly elevation change rate; hence, the DEM for each month from

November 2018 to November 2019 can be derived theoretically.



We calculated the mean elevation of the calculated grids at four resolutions according to different main basins (Figure 5). It can be clearly seen that with increasing resolution, the calculated elevations were mostly higher, while the calculated elevation were decreased only in basin 1 under 2 km resolution and in the GLA region under 1 km and 2 km resolution. The regions with the largest bias were concentrated in the low-latitude regions of Greenland, with a maximum deviation of more than 40 m. The elevation difference did not increase with varying resolutions for the GLA region. Because the topography here was more complex, it was possible that the elevation could be either overestimated or underestimated on different glaciers, and this uncertainty alleviated the elevation differences.

**4.2 ICESat-2 DEM uncertainty**

The uncertainty in the DEM shows an obvious spatial pattern. The uncertainty presents an increasing trend from the interior to the margins and from the north to the south. In the inner ice sheet, the uncertainty is approximately less than 0.5 m, but for the periphery of Greenland, higher uncertainty can be observed **(Figure 4 (b))**. The glaciers in northern Greenland exhibit uncertainties of 2-5 m, while the uncertainty increases to 10 m in southern Greenland. The generated slope and slope uncertainty are in good agreement with the elevation uncertainty, and the uncertainty is large at the edges, which is also concurrent with slopes exceeding 1° **(Figure 4(c) and (d))**. The error for the independent glaciers is due to the large fluctuation in the surface slope, and the ICESat-2 DEM with a resolution of 500 m has difficulty capturing detailed elevation patterns..

**5. Discussions**

**5.1 Comparison of ICESat-2 DEM with IceBridge data**

The elevations of ICESat-2 DEM and IceBridge show general agreement (Figure 6 and Table 2). Overall, the entire area of Greenland exhibits MED, MD, and MAD values of -0.48 m, -1.90 m, and 2.73 m, respectively, which compare favourably to those of IceBridge data. The MEDs do not differ in the calculated and interpolated grids (both -0.48 m). However, the performance decreases in the interpolated regions, which can be seen from the remaining parameters, such as MD, MAD and STD. The correlations between ICESat-2 DEM and IceBridge are high for all of Greenland, the calculated grids, and the interpolated grids, with the highest correlation occurring in all of Greenland and the calculated grids (both 0.9999). Kriging interpolation is used to fill the remaining 2% of the DEM gaps, which are concentrated in the southernmost part of Greenland. Although ICESat-2 has a higher coverage percentage than previous altimeters, interpolation errors caused by the data gaps cannot be avoided. Poorer performance in the interpolated grids is reasonable due to the low spatial correlation in the regions with large surface fluctuations. Although this accuracy is worse than that of the calculated grids, it still agrees well with that of the IceBridge data.

We also compared the accuracies according to the different basins. The data from May 2019 covered only 10 basins in Greenland (Table 3). Regionally, the MED of each basin ranges from -0.95 m to 0.20 m, with the largest deviation appearing



in basin 5. The MAD, STD, RMSE, and LE90 show the same trend, with the largest bias in the GLA region. The highest MAD is located in basin 5, excluding the GLA region. For the calculated grids, all basins display a difference of less than 5

m, but the difference in the GLA region can be up to 15 m. For the interpolated grids, the poorest performance still appears in basin 5 and the GLA region. Overall, the accuracy of the ICESat-2 DEM shows an apparent spatial trend. Better accuracy is observed in the north than in the south, mainly because the proportion of calculated grids in the southern basin is small, and the results of coarse resolution are mostly adopted. Additionally, due to higher temperature sensitivity, true elevation changes during the acquisition times of IceBridge and ICESat-2 lead to another part of the bias.

The ICESat-2 DEM has very high accuracy, but there are still some differences between the ICESat-2 DEM and IceBridge data, which are mainly due to the inconsistent resolutions of the two datasets. It should be noted that the IceBridge data used for evaluation were distributed only at latitudes below 75°N. The DEM was mostly derived by the results with a 1 km resolution or even with 2 km and 5 km resolutions. The ICESat-2 DEM should have higher accuracy in the regions with 500 m resolution (located north of 75°N). Hence, these biases are acceptable because the evaluated value represents the upper

bound of the ICESat-2 DEM bias, and the deviation should be smaller when considering Greenland as a whole.

**5.2 Comparison with other available DEMs**

We first compared the spatial coverage of different DEMs and found that the coverage percentage of all DEMs exceeded 99% (Table 1). Of these, the CryoSat-2 DEM has the lowest coverage (99.33%), and most peripheral glaciers show a large number of voids. TanDEM has the second lowest coverage (99.93%), and although the release product has filled the gaps,

there are still small void grids over peripheral Greenland due to radar shadow, echo delays and phase unwrapping errors. The ICESat DEM has the largest coverage (99.99%), which is due to its 5.5-20 km search radii for different latitudes when calculating the elevations of each grid. The ICESat-2 DEM coverage is comparable to that of the 500 m and 1 km ArcticDEM (99.98%).

**Figure 7** depicts the elevation differences between the new ICESat-2 DEM and the other five published available DEMs.

Larger elevation differences can be found in peripheral Greenland than those in all other DEMs, but the elevation differences in the interior ice sheets are much smaller. Significant positive patterns can be seen in the elevation difference between the ICESat-2 DEM and TanDEM on the Greenland ice sheet due to X-band penetration. All the difference maps show significant negative values in the Jakobshavn Isbrae glacier, which is the area experiencing the greatest loss of the Greenland ice sheet (Smith et al., 2020). This reflects the real elevation changes in Greenland at different DEM acquisition times. The

ICESat-2 DEM is generally close to that of the 500 m ArcticDEM except in complex terrains. As satellite optical images have a much finer original spatial resolution (2 m) than ICESat-2, larger elevation differences in high-slope regions are reasonable. However, these differences are much smaller than those of other DEMs, which proves the greater reliability of the ICESat-2 DEM.

We evaluated the vertical accuracies of all DEMs using IceBridge data as described in section 3.2. To ensure comparability

between datasets, we used only IceBridge data that overlapped with the corresponding DEM period for evaluation.





Compared with those of DEMs generated by altimeters, the accuracy of the DEM is greatly improved (Table 4). The ICESat-2 DEM generally performs the best. For all of Greenland, the Cryosat-2 DEM has the smallest MED of 0.03 m, and the ICESat-2 DEM ranks next. The ICESat-2 DEM has the best performance with regard to the rest of the parameters. The correlation can reach 0.9999 for all of Greenland. The ICESat-2 DEM shows the best performance in areas with elevations

greater than 2000 m. In areas with elevations less than 2000 m, the ICESat-2 DEM still shows the best performance, except that the smallest MED appears in the Cryosat-2 DEM. Contrary to expectations, there is no elevation underestimation in the CryoSat-2 DEM, possibly because slope and topographic corrections were performed in the DEM.

Compared with DEMs generated by image pairs, the ICESat-2 DEM also shows good performance (Table 4). Affected by the penetration of SAR signals into dry snow in high-elevation areas, TanDEM has the worst performance in the regions

with elevations above 2000 m, corresponding to an MED of -3.76 m. The MED (-2.32 m) is lower in the regions with elevations below 2000 m, indicating that TanDEM is less affected by penetration in the low-elevation regions. The snow penetration of the X-band is mainly concentrated in dry snow areas, and the penetration into wet snow in low-elevation areas is limited and can even be neglected. In addition, ICESat data alone were used to calibrate the raw DEMs in the coastal regions (Wessel et al., 2016), and the effect of penetration was alleviated to a certain extent. All parameters indicate that the

accuracy of the ICESat-2 DEM is higher than that of the 1 km ArcticDEM. The accuracy is highly comparable to that of the 500 m ArcticDEM. The MED is better for all of Greenland, and the rest of the parameters are similar. However, the ICESat-2 DEM has a higher elevation correlation with IceBridge, especially in regions with elevations greater than 2000 m.

The accuracy of all DEMs is affected by varying topography. All DEMs tend to increase in accuracy with increasing elevation, with the exception of TanDEM (Figure 8(a)). The ICESat-2 DEM is clearly superior in regions with elevations

less than 500 m compared with DEMs generated by other altimeters. It is also superior to the 1 km ArcticDEM at elevations below 2000 m, and the accuracy is close to that of the 500 m ArcticDEM. Similarly, elevation differences between each DEM and IceBridge were calculated with respect to the surface slope (Figure 8(b)). Similar to the elevation results, only TanDEM shows a trend of improvement in accuracy as the slope increases and has the highest accuracy in the high-slope (also the low-elevation) regions. The CryoSat-2 DEM, 1 km ArcticDEM and ICESat DEM all tend to show higher errors as

the slope increases, while both the ICESat-2 DEM and 500 m ArcticDEM maintain high accuracy and stability under all slope conditions. A trend of decreasing performance is found with increasing roughness for all DEMs except TanDEM (Figure 8(c)). The ICESat-2 DEM and 500 m ArcticDEM have the best performance, with deviations of less than -0.5 m for all roughnesses. In terms of aspect, DEMs generated by stereo pairs have obvious directivity (Figure 8(d)). The accuracy on the north slope is significantly lower. This is mainly due to the poor illumination condition of the images in the north

direction, which affects the accuracy of the generated DEM. The accuracy of satellite laser altimeters is affected by surface roughness, slope, and other environmental factors (Brunt et al., 2017). The measurement footprint is sensitive to changes in surface terrain, and a flatter surface provides a more uniform reflection than a steeper surface. A more accurate height measurement of the original ICESat-2 footprints can be obtained in the low-slope regions.



In summary, the ICESat-2 DEM is superior to the previous satellite altimeter-derived DEMs in both spatial resolution and elevation accuracy. Compared with the $6.9\times10^6$ footprints used in the ICESat DEM and the $7.5\times10^6$ footprints used in the CryoSat-2 DEM, approximately 80 times as many data were used to generate the DEM; thus, the ICESat-2 DEM with higher resolution of 500 m can be obtained. Due to the characteristics of ICESat-2 itself, the denser orbit density and higher measurement precision, higher elevation accuracy can be achieved. Smaller differences can be found when compared to the satellite image-derived DEMs. When comparing the DEMs generated by SAR interferometry, the ICESat-2 DEM has an increased accuracy because it is not affected by the penetration depth into snow. The penetration depth depends on the radar wavelength and the snow and ice properties, leading to a significant underestimation of elevation (Guerreiro et al., 2016). Although model-based or empirical models can correct the penetration bias to some extent (Abdullahi et al., 2019), such correction is generally restricted to the regional scope (Wessel et al., 2021), and there are still significant elevation underestimations. The ArcticDEM has high precision because ICESat data were used to calibrate the ArcticDEM in both the horizontal and vertical directions to increase the accuracy. However, the ICESat data predate the ArcticDEM by almost 10 years, and actual changes in the ice surface introduce additional uncertainties. Finally, there may also be systematic errors between the ArcticDEM's different sensors *(Candela, 2017)*.

The problem remains with the uncertainty the ignorance of the elevation changes during DEM data acquisition. The generation method of the ICESat-2 DEM is similar to that of the ICESat DEM, but the advantage of the former is that we took the data acquisition time of each point into account and used the smaller search radius in the interpolation, minimizing the time uncertainty as much as possible.

## 6. Conclusions

A new digital elevation model of Greenland was provided based on the ICESat-2 observations acquired from November 2018 to November 2019. A model-fit method was applied within the grid cells within different spatial resolutions to estimate the surface elevations with a modal resolution of 500 m. Different resolutions ensured that more ICESat-2 data could be utilized to reduce the number of interpolated grids, which contributed to less bias in the elevation estimation. Overall, we estimated the uncertainty with a median difference of -0.48 m for all of Greenland, which represents the upper bound of the ICESat-2 DEM bias. In addition, the accuracy of the ICESat-2 DEM shows an apparent spatial trend, and better accuracy can be observed in the north than in the southern basins, owing to the denser coverage of ICESat-2 tracks in the high-latitude regions.

Compared with other published Greenland DEMs, i.e., the ICESat DEM, CryoSat-2 DEM, 1 km ArcticDEM, 500 m ArcticDEM, and TanDEM, the ICESat-2 DEM shows great accuracy stability under various topographic conditions. Compared to DEMs derived from satellite altimeters, the ICESat-2 DEM has the finest spatial resolution for a Greenland DEM and hence has more accurate elevation measurements. Obvious elevation underestimation can be seen in the TanDEM due to penetration into snow. Compared to the ArcticDEM, the ICESat-2 DEM has smaller differences in the Greenland ice



sheet, which implies the reliability of the ICESat-2 DEM. Although the uncertainties in the ICESat-2 DEM are affected by the ICESat-2 measurements themselves and the coarse spatial resolution in the low-latitude regions, the advantage of accurate time can benefit studies of elevation change and mass balance in Greenland. With more ICESat-2 observations becoming available, more ICESat-2 data can be used to generate DEMs with higher resolution, especially in the
southernmost glaciers

## Data availability

The elevation and elevation uncertainty maps of Greenland can be downloaded from National Tibetan Plateau Data Center, Institute of Tibetan Plateau Research, Chinese Academy of Sciences at (https://data.tpdc.ac.cn/en/disallow/07497631-0475-48b5-ba53-c17f9076c72f/, Fan et al, 2021) during under review.

## Author contributions


Yubin Fan performed the DEM generation and wrote the manuscript; Chang-Qing Ke contributed to the conception of the study and supervised the work. Xiaoyi Shen contributed to the discussion and advised on the comparison with IceBridge data. All authors contributed to the discussion of the results and to the improvement of the manuscript.

## Competing interests

The authors declare that they have no conflict of interest.

## Acknowledgements

This work is supported by the Program for National Natural Science Foundation of China (grant No. 41830105). The ICESat-2 data were obtained from the National Snow and Ice Data Center (http://nsidc.org). We also thank the High Performance Computing Center, Nanjing University for the computing support.

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





**Figure 1: IceBridge data acquired in May 2019, which were used to evaluate the generated ICESat-2 DEM, covering regions in Greenland with various terrain conditions: (a) elevation, (b) slope, (c) aspect, and (d) roughness. Labels in the picture are the main glaciers in Greenland.**

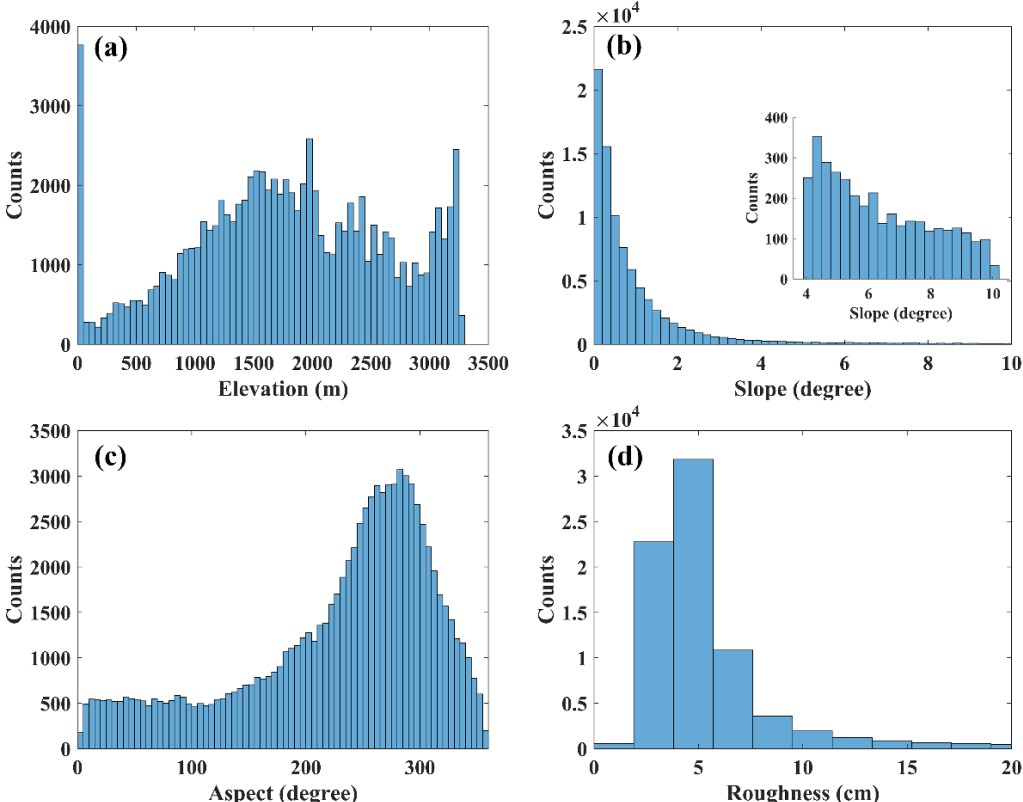

**Figure 2: Histogram of the (a) surface height, (b) surface slope, (c) aspect, and (d) roughness derived from IceBridge data. The inset figure shows the histogram of surface slope for values between 4° and 10°.**



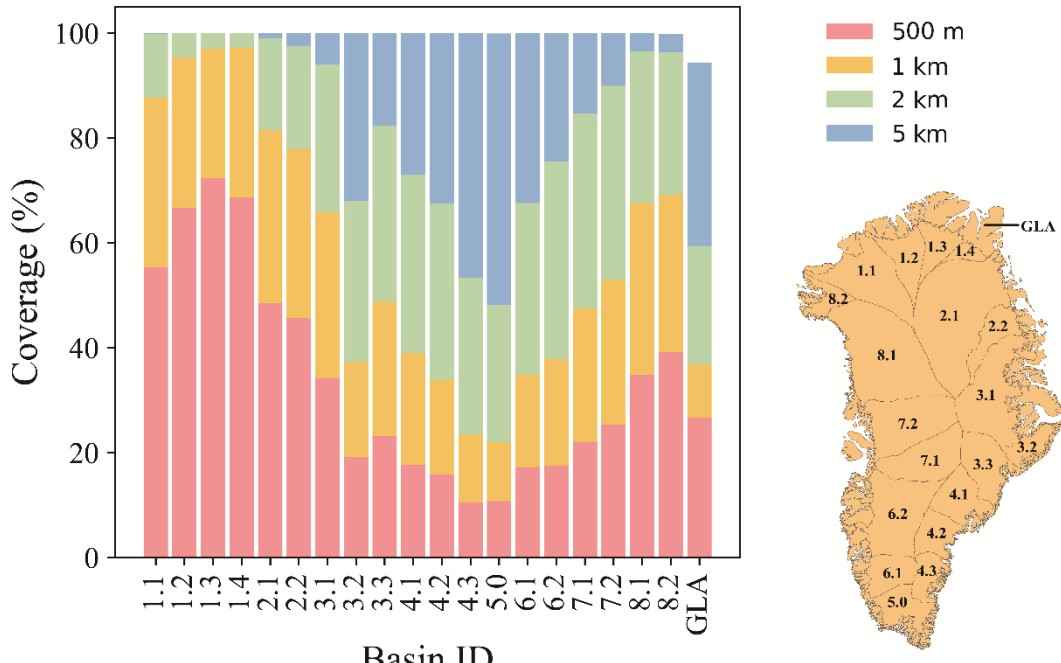


**Figure 3: Coverage percentages of calculated elevation grids by ICESat-2 observations of 500 m, 1 km, 2 km, and 5 km. The basin boundaries are from** Zwally et al. (2002)**, which divides Greenland into 8 main basins, covering approximately 1.72×10⁶ km². Ice caps and glaciers that are not connected with the ice sheet are marked as GLA.**

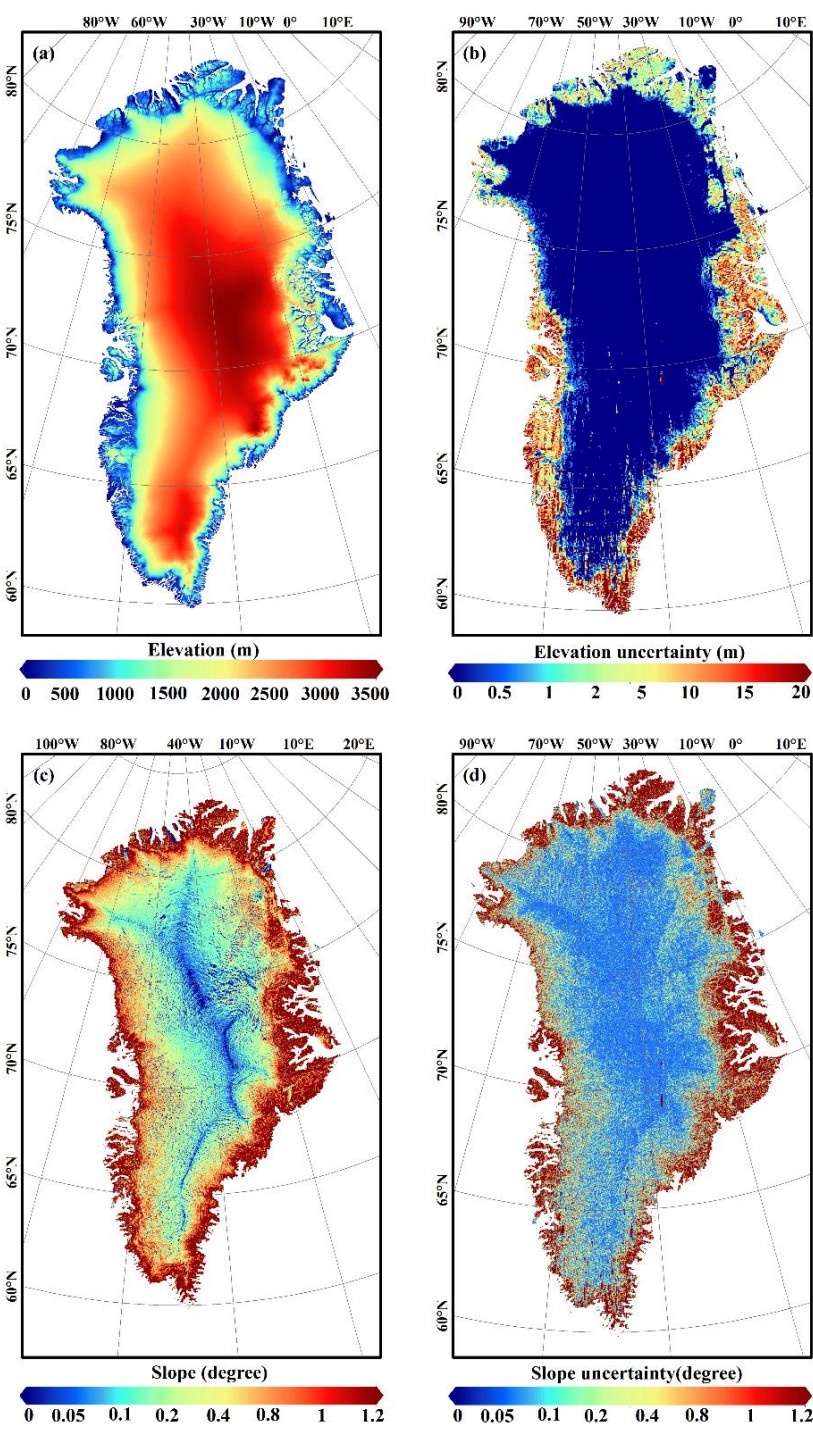

Figure 4: (a) Elevation of the Greenland DEM calculated from 13 months of ICESat-2 footprints acquired between November 2018 and November 2019. (b) The elevation uncertainties are the 95% confidence intervals for elevations in the calculated grids, and the kriging variance errors are the uncertainties in the interpolated grids. (c) Slope and (d) slope uncertainty derived from the elevation map (a).

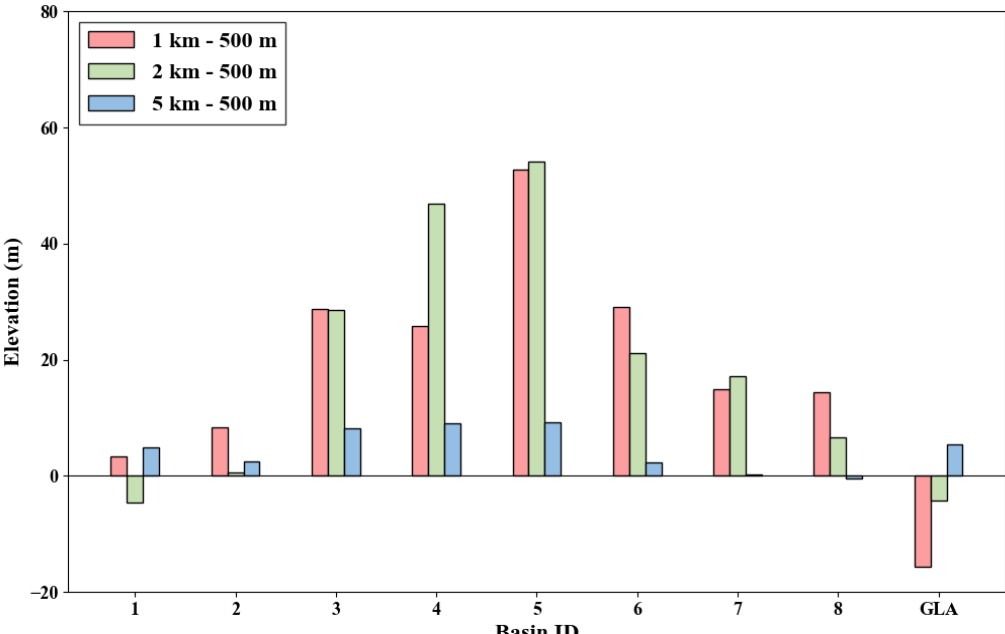

**Figure 5: Elevation differences of 9 main regions under different resolutions, which are calculated by subtracting the 500 m DEM from the 1 km DEM, 2 km DEM and 5 km DEM through the overlapping grids of different DEMs. The colour bar shows the mean elevation differences of these regions.**

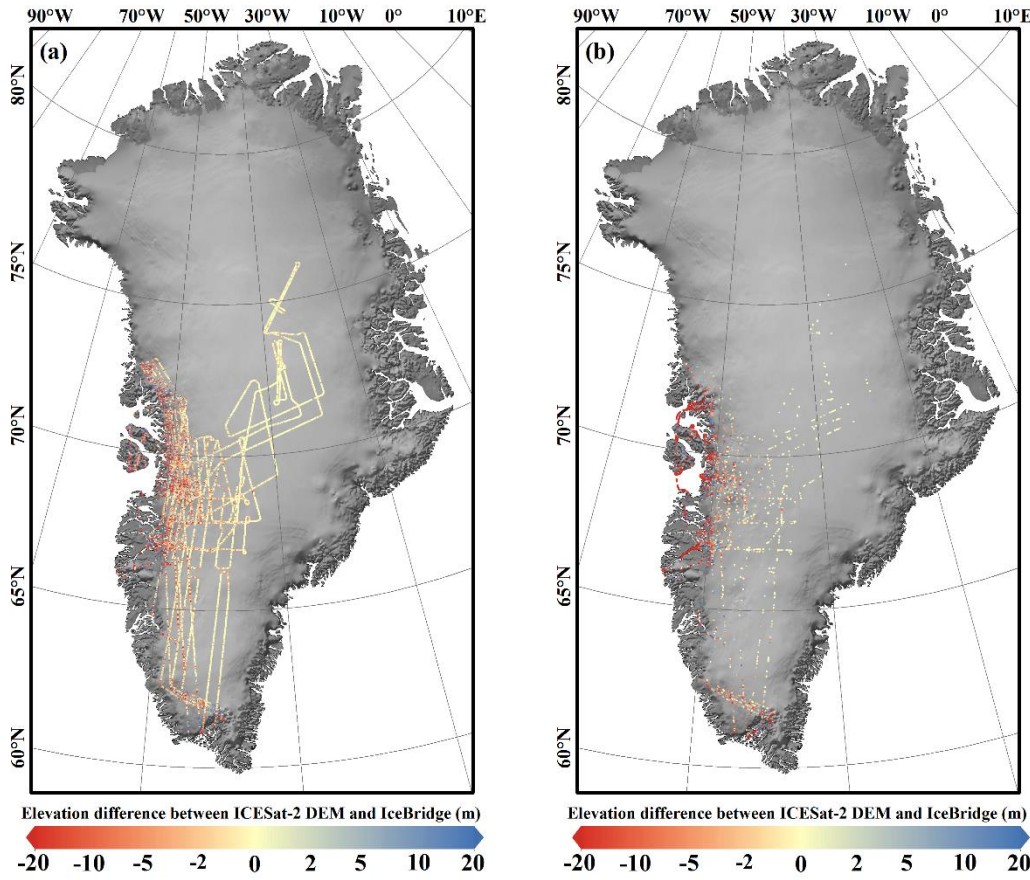

**Figure 6: Elevation difference calculated as IceBridge data subtracted from the new ICESat-2 DEM. (a) Calculated grids and (b)**
**465  interpolated grids. IceBridge data were acquired in May 2019.**





**Figure 7: Elevation differences calculated between the new ICESat-2 DEM and the other five published available DEMs. (a) ICESat DEM, (b) CryoSat-2 DEM, (c) 500 m ArcticDEM, (d) 1 km ArcticDEM, and (e) TanDEM. For each picture, the previously published DEM was resampled to 500 m, and the difference was calculated as the resampled DEM subtracted from the new ICESat-2 DEM.**




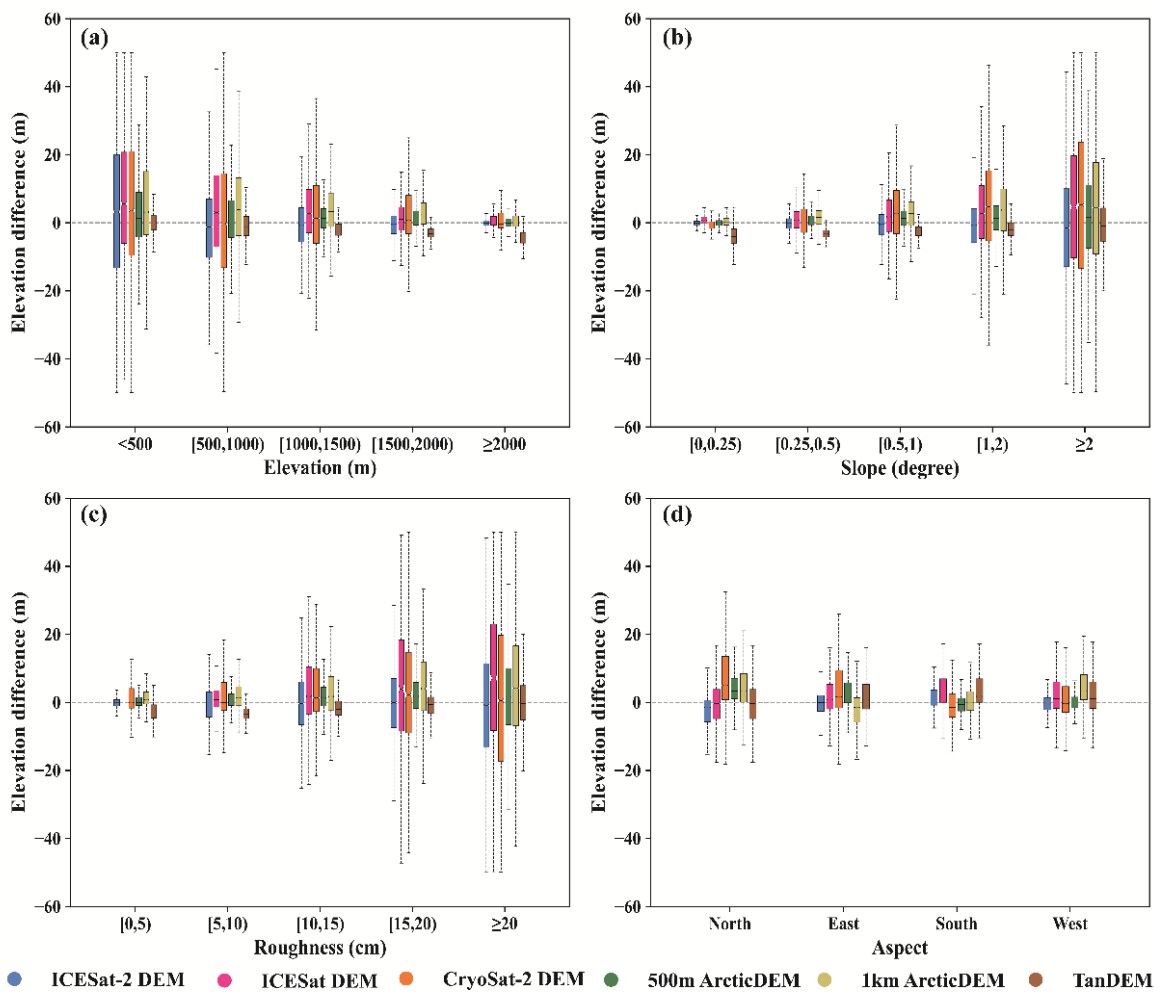

**Figure 8: Elevation differences between different DEMs and IceBridge data under different terrain conditions. (a) Elevation, (b) slope, (c) roughness, and (d) aspect. The solid black lines near the box centres denote median values of elevation differences, the upper and lower boundaries of each box denote upper and lower quartiles (Q1 and Q3), the length means the interquartile range (IQR), and the top and bottom lines denote the range [Q1-1.5 IQR~Q3+1.5 IQR].**




**Table 1: Published Greenland DEMs used in this study. Note that the ArcticDEM has higher resolutions of 2, 10, 32, and 100 m and that we used only resolutions of 500 and 1000 m for comparison.**

| DEM | Data sources | Spatial coverage | Temporal coverage | Resolution (m) | DEM generation method |
|---|---|---|---|---|---|
| ICESat DEM | ICESat | 99.99% | 2003-2005 | 1000 | bi-quadratic surface fit |
| ArcticDEM | GeoEye-1, WorldView-1, WorldView-2, WorldView-3 | 99.96% (500m), 99.98% (1000m) | 2015-2018 | 500,1000 | stereopair, calibrated by ICESat |
| TanDEM DEM | TanDEM-X, TerraSAR-X | 99.93% | 2011-2014 | 90 | radar interferometry |
| CryoSat-2 DEM | CryoSat-2 AWI L2 data | 99.33% | 2011-2014 | 1000 | Original kriging interpolation |


**Table 2: Elevation differences between the ICESat-2 DEM and IceBridge data for all of Greenland and the calculated and interpolated grids. IceBridge data were acquired in May 2019.**

| Region (grid numbers) | MED(m) | MD(m) | MAD(m) | STD(m) | RMSE(m) | LE$_{90}$(m) | R |
|---|---|---|---|---|---|---|---|
| Total (70046) | -0.48 | -1.90 | 2.73 | 11.31 | 11.47 | 18.61 | 0.9999 |
| Calculated grids (61506) | -0.48 | -0.77 | 2.39 | 10.34 | 10.54 | 17.01 | 0.9999 |
| Interpolated grids (8540) | -0.48 | -2.06 | 7.57 | 16.66 | 16.68 | 27.41 | 0.9998 |



**Table 3: Elevation differences between the ICESat-2 DEM and IceBridge data in different basins, calculated for all of Greenland and the calculated and interpolated grids. Basins with fewer than 30 grids were excluded. IceBridge data were acquired in May 2019.**

| Basin | Region (grid numbers) | MED (m) | MD (m) | MAD (m) | STD(m) | RMSE(m) | LE$_{90}$(m) | R |
|---|---|---|---|---|---|---|---|---|
| 2.1 | Total (4570) | 0.20 | -0.14 | 0.72 | 2.52 | 2.53 | 4.15 | 0.9999 |
| | Calculated (4525) | 0.20 | -0.15 | 0.71 | 2.53 | 2.53 | 4.16 | 0.9999 |
| | Interpolated (45) | 0.74 | 1.09 | 0.89 | 1.62 | 1.93 | 2.66 | 0.9999 |
| 3.1 | Total (2171) | 0.03 | 0.10 | 0.50 | 1.80 | 1.80 | 2.96 | 0.9997 |
| | Calculated (2142) | 0.03 | 0.10 | 0.50 | 1.81 | 1.81 | 2.98 | 0.9997 |
| | Interpolated (29) | -0.16 | -0.22 | 0.27 | 0.58 | 0.62 | 0.96 | 1.0000 |
| 4.2 | Total (189) | 0.12 | 0.12 | 0.40 | 1.44 | 1.44 | 2.36 | 0.9992 |
| | Calculated (158) | 0.16 | 0.16 | 0.40 | 1.53 | 1.54 | 2.52 | 0.9991 |
| | Interpolated (31) | 0.04 | -0.05 | 0.31 | 0.76 | 0.74 | 1.24 | 0.9998 |
| 5 | Total (4965) | -0.95 | -0.22 | 5.11 | 12.92 | 12.92 | 21.26 | 0.9996 |
| | Calculated (3438) | -0.78 | 0.67 | 4.09 | 12.37 | 12.38 | 20.34 | 0.9997 |
| | Interpolated (1527) | -2.04 | -2.21 | 7.35 | 13.90 | 14.07 | 22.86 | 0.9994 |
| 6.1 | Total (4999) | -0.34 | -1.32 | 2.06 | 8.79 | 8.89 | 14.46 | 0.9998 |
| | Calculated (4621) | -0.34 | -1.58 | 1.93 | 7.71 | 7.87 | 12.68 | 0.9999 |
| | Interpolated (378) | -0.38 | 1.86 | 6.94 | 16.92 | 16.99 | 27.82 | 0.9997 |
| 6.2 | Total (25560) | -0.87 | -2.87 | 2.92 | 9.78 | 10.19 | 16.08 | 0.9999 |
| | Calculated (22227) | -0.91 | -3.18 | 2.64 | 8.95 | 9.50 | 14.73 | 0.9999 |
| | Interpolated (3333) | -0.33 | -0.75 | 5.98 | 13.90 | 13.92 | 22.87 | 0.9998 |
| 7.1 | Total (15867) | -0.63 | -1.93 | 2.61 | 9.69 | 9.88 | 15.94 | 0.9999 |
| | Calculated (14046) | -0.65 | -2.07 | 2.44 | 9.24 | 9.47 | 15.19 | 0.9999 |
| | Interpolated (1821) | -0.42 | -0.85 | 4.82 | 12.62 | 12.64 | 20.76 | 0.9999 |
| 7.2 | Total (19092) | -0.53 | -2.01 | 2.93 | 11.51 | 11.68 | 18.93 | 0.9999 |
| | Calculated (17513) | -0.54 | -2.15 | 2.67 | 10.98 | 11.19 | 18.06 | 0.9999 |
| | Interpolated (1579) | -0.35 | -0.41 | 8.00 | 16.18 | 16.18 | 26.62 | 0.9998 |
| 8.1 | Total (1698) | -0.87 | -4.34 | 4.32 | 13.21 | 13.90 | 21.72 | 0.9999 |
| | Calculated (1635) | -0.89 | -4.49 | 4.21 | 13.20 | 13.94 | 21.72 | 0.9999 |
| | Interpolated (63) | -0.16 | -0.34 | 7.41 | 12.76 | 12.66 | 20.98 | 0.9999 |
| GLA | Total (4891) | -0.42 | -0.77 | 17.15 | 24.14 | 24.15 | 39.71 | 0.9984 |
| | Calculated (3254) | -0.31 | -0.70 | 14.79 | 22.54 | 22.55 | 37.08 | 0.9987 |
| | Interpolated (1637) | -0.90 | -0.92 | 22.17 | 27.05 | 27.06 | 44.49 | 0.9970 |





**Table 4: Elevation differences of the ICESat-2 DEM and other published DEMs with respect to IceBridge data. All of Greenland and regions with elevations above 2000 m and below 2000 m were compared.**

| | DEM (grid numbers) | MED (m) | MD (m) | MAD(m) | STD(m) | RMSE(m) | LE$_{90}$(m) | R |
|---|---|---|---|---|---|---|---|---|
| all Greenland | ICESat-2 DEM (70046) | -0.48 | -1.90 | 2.73 | 11.31 | 11.47 | 18.61 | 0.9999 |
| | ICESat DEM (32106) | 1.02 | 2.15 | 3.66 | 13.22 | 13.40 | 21.75 | 0.9947 |
| | CryoSat-2 DEM (113538) | 0.03 | 2.52 | 4.07 | 12.82 | 13.07 | 21.09 | 0.9679 |
| | 500 m ArcticDEM (346043) | 0.49 | 1.13 | 2.08 | 8.56 | 8.63 | 14.08 | 0.9994 |
| | 1 km ArcticDEM (151558) | 1.46 | 2.53 | 3.48 | 10.98 | 11.27 | 18.06 | 0.9988 |
| | TanDEM (418676) | -2.75 | -1.78 | 3.19 | 6.33 | 6.58 | 10.41 | 0.9999 |
| Elevation above 2000 m | ICESat-2 DEM (28321) | -0.23 | -0.63 | 0.97 | 4.63 | 4.67 | 7.61 | 0.9999 |
| | ICESat DEM (11210) | 0.64 | 0.51 | 1.40 | 5.99 | 6.01 | 9.85 | 0.9979 |
| | CryoSat-2 DEM (46908) | -0.36 | 1.92 | 1.82 | 8.37 | 8.59 | 13.77 | 0.9873 |
| | 500 m ArcticDEM (114165) | -0.07 | 0.40 | 0.97 | 3.26 | 3.28 | 5.36 | 0.9932 |
| | 1 km ArcticDEM (52155) | 0.33 | 0.32 | 1.52 | 5.27 | 5.28 | 8.67 | 0.9904 |
| | TanDEM (83733) | -3.76 | -3.76 | 3.79 | 2.49 | 4.51 | 4.10 | 0.9999 |
| Elevation below 2000 m | ICESat-2 DEM (41725) | -1.32 | -2.77 | 5.41 | 14.09 | 14.36 | 23.17 | 0.9996 |
| | ICESat DEM (20896) | 1.97 | 3.03 | 6.49 | 15.72 | 16.01 | 25.86 | 0.9845 |
| | CryoSat-2 DEM (66630) | 0.98 | 2.94 | 6.84 | 15.18 | 15.46 | 24.97 | 0.9062 |
| | 500 m ArcticDEM (231878) | 1.16 | 1.49 | 3.13 | 10.18 | 10.29 | 16.75 | 0.9992 |
| | 1 km ArcticDEM (99403) | 2.99 | 3.70 | 5.39 | 12.86 | 13.38 | 21.15 | 0.9980 |
| | TanDEM (334943) | -2.32 | -1.29 | 3.00 | 6.88 | 7.00 | 11.32 | 0.9998 |