# Peer review of "A new Greenland digital elevation model derived from ICESat-2"

_Earth System Science Data, 2021_

## Referee Comment (RC2)

DEMs are the basic datasets for digital hydrology, watershed analysis, quantification of remote sensing, glacier change, etc. Compared with the optical stereo images (photogrammetric method) and radar interferometry (microwave band), the DEM data obtained by laser altimetry satellite data has higher reliability, especially for the surface elevation of glaciers and snow cover. In this paper, a new Greenland DEM is derived from ICESAT-2 withe a definite time (13 months), which is very meaningful and practical. However, the paper still has many problems, and substantial revisions are required before the acceptance of this manuscript can be recommended.

Specific comments:
1. There are some problems with cryosat-2 satellite orbit description in the article. Some parameter values have inconsistencies or errors, please check carefully. For example, " which is a great improvement over CryoSat-2's along-track distance of 1.5 km and cross-track distance of 3 km" .

2. Each parameter in the formula needs to be defined, but there are no explanations for many parameters in the paper, such as formula (1)-(4). In formula (5), what does "h" mean and how to input its parameter values?

3. There is no need to write specific formulas for commonly used parameters in the paper, such as slope and aspect.

4. How to use ICESAT-2 laser point cloud data to simulate DEM data at 500 m grid scale is the focus of this paper, but the paper does not describe it clearly. How to achieve "To improve ICESat-2 data utilization, DEMs with 1 km and 2 km resolution across all of Greenland and an additional 5 km resolution in southernmost Greenland were used to fill the DEM gaps. Kriging interpolation was used to 15 fill the remaining 2% of void grids that were insufficiently observed by ICESat-2 measurements." And how to get calculated grids and interpolated grids in Figure 6. Therefore, it is recommended to expand the content of section 3.1.

5. "We set the minimum number of grid points to 10 and the minimum timestamp to 2 months", are you sure it is 2 months timestamp here? Generally speaking, the revisit period of icesat-2 is 91 days.

6. Some descriptions in the paper are not clear. For example, in Figure 4(b) and (d), how is the elevation uncertainty calculated and which index is used?

7. In general, the labels of the figures are arranged in the paper from small to large, please check the order of Figure 4 and Figure 5.

8. The structure of the conclusion and discussion section is a bit confusing, please reorganize it. Generally speaking, accuracy verification is part of the results.

---

## Author Comment (AC1)

Reviewers' Comments:
Reviewer: #1
This study utilizes ICESat-2 data from November 2018 to November 2019 to generate a new DEM over Greenland, and validates the newly generated DEM with IceBridge ATM data obtained in May 2019. This study also presents a comparison of different DEM products. I think that this paper is interesting and the topic is suitable to ESSD. However, several major comments should be seriously considered.

We thank you for the helpful feedback, these suggestions have significantly improved the text and figures, we are appreciative of your help and time.

Major comments:

1. The present language quality is not good enough and needs to be improved throughout. For example, Page 1 Line 10-11, "but long temporal coverage introduced additional time uncertainty to scientific research", what does time uncertainty mean? Line 11-12 "with a definite time", what does it mean? Page 2 Line 49, "Hence"? Page 3 Line 97, "by different beams"? Page 9 Line 253, "Of these" should be "Among them".

Responses: We have changed the structure of the article and some statements in the article, and these parts are shown in red in the revised version.

(a) Page 1 Line 10-11, "but long temporal coverage introduced additional time uncertainty to scientific research", what does time uncertainty mean?
Responses: Time uncertainty refers to temporal resolutions. The long timespan of applied source data leads to the low temporal resolutions of previous DEMs. When it was applied to calculate elevation and mass changes, it is hard to quantify the years when these changes occurred.

Line 11-12 "with a definite time", what does it mean?
With a definite time means that ICESat-2 DEM has a specific time-stamp (e.g May 2019).

(b) Page 2 Line 49, "Hence"?
Responses: We changed the sentence to 'In addition, owing to the wide coverage (86°N-86°S), high single-point accuracy (0.1-0.15 m), and small footprint size (70 m) (Zwally et al., 2002), ICESat has the ability to measure the elevation of entire Greenland.'

(c) Page 3 Line 97, "by different beams"?
Responses: We changed the sentences to 'Hence, we included weak beams to increase spatial coverage and data point utilization due to no systematic errors were found in strong and weak beams in ICESat-2 elevation measurements.'

(d) Page 9 Line 253, "Of these" should be "Among them"

Responses: Accept and revised.

2. The resample resolution of IceBridge is about 25 m, how to validate the ICESat-2 DEM with different spatial resolution (namely, 500 m, 1 km, 2 km). Please clarify.

Responses:
One ICESat-2 DEM grid cell usually has several IceBridge measurement points. In each grid cell, the ICESat-2 DEM elevation values were subtracted from the median of all IceBridge elevations within it, and this difference was seen as the final bias of the corresponding cell.
The final posted DEM is a composite of Greenland DEMs with different resolutions, so we also compared the elevation differences between the ICESat-2 DEM and IceBridge data of grid cells originated from 500m, 1km (not covered by 500m DEM) and 2km (not covered by 500m and 1km DEMs), respectively (Table 3) as the method above.

Table 3: Elevation differences between the ICESat-2 DEM under different DEM resolutions and IceBridge data.

| DEM resolution (grid numbers) | MED(m) | MD(m) | MAD(m) | STD(m) | RMSE(m) | R |
|---|---|---|---|---|---|---|
| 500m (11186) | -0.09 | -0.16 | 0.60 | 2.55 | 2.55 | 0.9999 |
| 1km (6903) | -0.01 | -0.04 | 0.71 | 2.81 | 2.81 | 0.9999 |
| 2km (8453) | -1.37 | -1.78 | 2.52 | 6.34 | 6.59 | 0.9998 |

The results show that the DEM under 500m and 1km resolutions exhibit higher performances than that of 2 km-resolution DEM, and all biases of the three resolutions are smaller than the interpolation error.

3. Please rewrite the Method part in Page 6, Line 164-173, which is different to understand. When the ICESat-2 data were gridded to fine resolution (i.e. 500 m), there would be many gaps. These gaps will be filled with values from coarse-grid data (i.e. 1 km or 2 km)?
Responses: We rewrote the Method part as follows.

Firstly, Greenland DEM in four resolutions (500 m, 1 km, 2 km, and 5 km) were acquired by the spatiotemporal model fit process. However, these four types of DEM all include voids area thus we need to incorporate them to obtain final Greenland DEM results with the minimal gaps. We used Greenland DEM with 500 m resolution as our first DEM source. Afterwards, Greenland DEMs with 1 km, 2 km, and 5 km resolution were resampled to 500m by applying a bilinear method to fill the gaps in this DEM and the finer resolution as our first option.

"We set the minimum number of grid points to 10 and the minimum timestamp to 2

months ..."? What does this sentence mean and how the values of 10 and 2 were determined? "In addition, we introduced thresholds to remove outliers, which are RMSE≥10 m, the uncertainty of elevation change ≥10 m ..." Please clarify how these thresholds were calculated?

Responses: The minimum number of points in the grid is 7, because the quadratic function based on the local surface terrain used in the text has 7 unknowns. We set the minimum number of points as 10 to ensure the quality of the least square fitting. The minimum number of months is 2 months is also used to make sure the model fit can derive a result. If there is only one month's data in one grid, the fitting equation will have an infinite number of solutions.

We assumed that the maximum elevation change is 10 m/yr and its uncertainty is impossible to exceed 0.4 m/yr as Slater et al. (2018). Furthermore, we assumed that DEM uncertainty is less than 10 m and the maximum RMSE in each grid is 10 m. After this filter procedure, the elevation range is feasible since it is within the elevation range of published Greenland DEM products.

The above statement has also been added into the manuscript.

References: Slater T, Shepherd A, McMillan M, et al. A new digital elevation model of Antarctica derived from CryoSat-2 altimetry. The Cryosphere, 2018, 12(4): 1551-1562.

4. Please rewrite the conclusions in Section 5.2 and Section 6. Due to time discrepancy between different DEM products, I don't think it is possible to validate ICESat/GLAS DEM (2003~2005), ArcticDEM (2015~2018), TanDEM (2011~2014), CryoSat-2 DEM (2011~2014) using IceBridge data acquired in May 2019 and current results can support the conclusion that the ICESat-2 DEM showed significant improvements in accuracy compared with other altimeter-derived DEMs (in Page 1 Line 20~25). If possible, I suggest selecting areas with little elevation changes and doing the comparison.

Responses: We are sorry that our statement caused misunderstanding.

The data to evaluate other DEMs are the spatiotemporally matched IceBridge data. The IceBridge data to evaluate ArcticDEM were during 2015~2018, TanDEM during 2011~2014, CryoSat-2 DEM during 2011~2014, and the IceBridge data to evaluate ICESat/GLAS DEM were from 2009 since no data can be found in 2003~2005, so we used the data from the nearest year instead.

As you suggest, we selected areas with little elevation changes (-0.05~0.05 m/yr) (Smith et al., 2020) and did a further comparison.

| DEM (grid numbers) | MED(m) | MD(m) | MAD(m) | STD(m) | RMSE(m) | R |
|---|---|---|---|---|---|---|
| ICESat-2 DEM (15983) | -0.20 | -0.43 | 0.74 | 3.05 | 3.08 | 0.9999 |
| ICESat DEM (6903) | 0.63 | 0.35 | 1.10 | 4.15 | 4.16 | 0.9999 |
| CryoSat-2 DEM (27268) | -0.63 | 0.87 | 1.42 | 6.05 | 6.11 | 0.9999 |

| | | | | | | |
|---|---|---|---|---|---|---|
| 500 m ArcticDEM (54235) | -0.14 | 0.04 | 0.77 | 2.19 | 2.19 | 0.9999 |
| 1 km ArcticDEM (25675) | -0.04 | 0.06 | 0.82 | 2.67 | 2.67 | 0.9999 |
| TanDEM (50656) | -4.26 | -4.43 | 4.26 | 1.97 | 4.85 | 0.9999 |

The conclusion still stands that the ICESat-2 DEM showed significant improvements in accuracy compared with other altimeter-derived DEMs in areas with little elevation changes. The performance is also comparable to the stereo-photogrammetry-derived DEMs and is better than TanDEM.

The above statement has also been added into the manuscript.

Reference: Smith, B., Fricker, H. A., Gardner, A. S., et al. Pervasive ice sheet mass loss reflects competing ocean and atmosphere processes, Science, 368, 1239-+, https://doi.org/10.1126/science.aaz5845, 2020.

5. Since the ICESat-2 data is available from 2018 to present, two years' DEM products could be generated and compared. Otherwise, the specific time should be added to the title, for example, "A new Greenland digital elevation model derived from ICESat-2 during 2018-2019".

Responses: Accept and revised.

General comments:

Page 2 Line 37: "The previously published Greenland DEM dates back to the 1980s ...". "previously" would be "first", since "previously published" would have several DEM products?

Responses: Accept and revised.

Page 2 Line 38-40: "However, the data acquisition was limited by the low-visibility contrast between snow and ice surfaces (Noh and Howat, 2015), which introduced large time uncertainty into the DEM." How the low-visibility contrast introduced large time uncertainty. Please clarify.

Responses: We meant the low-visibility contrast may introduce large elevation uncertainty here. We rewrote the sentence as 'However, the low-visibility contrast between snow and ice surfaces may affect the radiometric and geometric quality of stereoscopic DEMs (Noh and Howat, 2015), which may introduce considerable uncertainty to the elevation.'

Page 3 Line 78-79: "IceBridge data were used to evaluate the accuracy for all of Greenland and for different basins."

According to Figure 1 in Page 16, the IceBridge data didn't coverage all the Greenland, please rewrite this sentence.

Responses: We rewrote this sentence to 'The overall accuracy of ICESat-2 DEM was evaluated by comparing to the spatiotemporally matched IceBridge data.'.

Page 3 Line 91-92: "However, for strong and weak beams in the ATL06 product, both beams in one pair show similar performance, with a median difference of -0.08 cm and -0.13 cm for strong beam2 and weak beam1"

The statement is confusing. "a median difference", compared with what? The results is for strong beam2 and weak beam1, what about the other beams? Please clarify.

Responses: The median difference is compared with contemporaneous IceBridge data. The median difference between strong beam4 and weak beam3 are -0.07m and -0.08m, and the median differences of the strong beam6 and the weak beam5 are the same, -0.03m. Beam2 and beam1 have the largest difference.

We also concluded the result in the text to 'Brunt et al. (2019) compared the elevation of ICESat-2 ATL06 product and GPS data, and found that the accuracy differences of strong and weak beams are less than 2cm. Shen et al. (2021) compared ICESat-2 ATL06 product with IceBridge data under complex terrain, and the result indicated that the height difference between them is also trivial. Hence, we included weak beams to increase spatial coverage and data point utilization due to no systematic errors were found in strong and weak beams in ICESat-2 elevation measurements.' to clarify the statement.

Reference:
Brunt, K. M., Neumann, T. A., Smith, B. E. (2019). Assessment of ICESat-2 ice sheet surface heights, based on comparisons over the interior of the Antarctic ice sheet. Geophysical Research Letters, 46, 13,072–13,078. https://doi.org/10.1029/2019GL084886.
Shen, X. Y., Ke, C. Q., Yu, X. N., et al. Int. J. Remote Sens., 42, 2556-2573, https://doi.org/10.1080/01431161.2020.1856962, 2021.

Page 4 Line 107-112, what do α, β, αs,n, αw,e, β0 stand for? Please clarify.
Responses: The calculation methods of slope and aspect have been deleted as the suggestion of reviewer#2, the detailed calculation can be referred from Shen et al. (2021). α is the slope, β is the aspect, $\alpha_{s,n}$ is the south-to-north slope, $\alpha_{w,e}$ is the west-to-east slope, $\beta_0$ stands for the aspect in degree value in the original text.

Reference: Shen, X. Y., Ke, C. Q., Yu, X. N., et al. Int. J. Remote Sens., 42, 2556-2573, https://doi.org/10.1080/01431161.2020.1856962, 2021.

Page 5 Line 145-150, what does h, a0 to a4 stand for? Please clarify how to get a0 to a4 and dh/dt.

Responses: $h_i$ means the elevation of each ICESat-2 footprint in one grid, h means the modelled elevation of the grid. $a_0$ to $a_4$ stand for surface elevation fluctuations (the fitting coefficients of a two-dimensional surface) and term t stands for seasonal changes.

All the coefficients were retrieved from an iterative least-squares fit to the observations in each grid. t is the month difference between May 2019 and ICESat-2 acquisition time, which adds a term of time, so the monthly elevation change can be derived.

The above statement has also been added into the manuscript.

---

## Author Comment (AC2)

Reviewer: #2

DEMs are the basic datasets for digital hydrology, watershed analysis, quantification of remote sensing, glacier change, etc. Compared with the optical stereo images (photogrammetric method) and radar interferometry (microwave band), the DEM data obtained by laser altimetry satellite data has higher reliability, especially for the surface elevation of glaciers and snow cover. In this paper, a new Greenland DEM is derived from ICESAT-2 with a definite time (13 months), which is very meaningful and practical. However, the paper still has many problems, and substantial revisions are required before the acceptance of this manuscript can be recommended.

We thank you for the helpful feedback, these suggestions have significantly improved the text and figures, we are appreciative of your help and time.

Specific comments:

1. There are some problems with cryosat-2 satellite orbit description in the article. Some parameter values have inconsistencies or errors, please check carefully. For example, "which is a great improvement over CryoSat-2's along-track distance of 1.5 km and cross-track distance of 3 km".

Responses: We have checked the CryoSat-2 satellite orbit description, and changed the sentence to 'A much finer observation can be obtained owing to its along-track distance of 0.7 m and cross-track distance of 3.3 km, which is a significant improvement compared with CryoSat-2's along-track distance of 0.3 km and cross-track distance of 1.5 km'

2. Each parameter in the formula needs to be defined, but there are no explanations for many parameters in the paper, such as formula (1)-(4). In formula (5), what does "h" mean and how to input its parameter values?

Responses: We have removed the formula (1)-(4) of slope and aspect calculation in the text as you suggest in comment #3. 'h' in the original formula 5 (now formula 1) means the modelled elevation, and $h_i$ is the elevations from ICESat-2 measurement points in one grid. The 'h' in one grid was calculated by performing an iterative least-squares fit model using all ICESat-2 measurements in this grid.

The above statement has also been added into the manuscript.

3. There is no need to write specific formulas for commonly used parameters in the paper, such as slope and aspect.

Responses: We have removed the calculated formulas of slope and aspect in the text as you suggest.

4. How to use ICESAT-2 laser point cloud data to simulate DEM data at 500m grid scale is the focus of this paper, but the paper does not describe it clearly. How to achieve "To improve ICESat-2 data utilization, DEMs with 1 km and 2 km resolution across all of Greenland and an additional 5 km resolution in southernmost Greenland were used to

fill the DEM gaps. Kriging interpolation was used to fill the remaining 2% of void grids that were insufficiently observed by ICESat-2 measurements."

Responses: We expanded the DEM simulation description in section 3.1, the rewritten texts are as follows 'After the aforementioned process, we have acquired Greenland DEM in four resolutions (500 m, 1 km, 2 km, and 5 km). However, these four types of DEM all include void areas thus we need to incorporate them to obtain final Greenland DEM results with the minimal gaps. Firstly, we used Greenland DEM with 500 m resolution as our primary DEM source. Afterwards, Greenland DEMs with 1 km, 2 km, and 5 km resolution were resampled to 500 m by applying a bilinear method to fill the gaps in this DEM and the finer resolution as our first option. Unavoidably, there are still some voids in the final Greenland DEM, but this has a minor impact on DEM accuracy.

   And how to get calculated grids and interpolated grids in Figure 6. Therefore, it is recommended to expand the content of section 3.1.
Responses: In this study, we described the unvoided area (98%) in the final Greenland DEM as 'calculated grids' and termed the rest (2%) as 'interpolated grids'. For the rest, an ordinary kriging approach was used to interpolate. The ICESat-2 DEM was posted at the modal resolution of 500 m after gap filling and interpolation. '.

5."We set the minimum number of grid points to 10 and the minimum timestamp to 2 months", are you sure it is 2 months timestamp here? Generally speaking, the revisit period of icesat-2 is 91 days.

Responses: One grid may contain several tracks, especially at high latitudes. The acquisitions time difference of these tracks is uncertain, and 91 days is the revisit cycle of one orbit. 2 months is the minimum requirement to solve the term dh/dt in the model fit.
Taking one 500m grid as an example, this grid contains 3 tracks, the left track (red one) was acquired in December,2018, and the right track (blue one) was acquired in January, 2019. These two tracks came from different Ground Reference Tracks of ICESat-2.

[Figure]

6.Some descriptions in the paper are not clear. For example, in Figure 4(b) and (d), how is the elevation uncertainty calculated and which index is used?

Responses: For the calculated grids, the regress function in MATLAB can return a matrix 'bint' that gives the range corresponding coefficient will be in with 95% confidence intervals, and the elevation uncertainty was described as equation below. For interpolated grid uncertainty estimation, we just used kriging variance error calculated by ArcGIS 10.6. There is a 95.5 percent probability that the actual elevation at the grid is the predicted raster value ± two times the square root of the variance error of the corresponding cell by assuming the kriging errors are normally distributed. Hence, the two times the square root of the value in the variance error was taken as the elevation uncertainty in the interpolated grids.

$$\text{elevation uncertainty}_{\text{calculated grids}} = t\,(1\text{-}0.025,\, n\text{-}p)\ \times SE\,(b_i)$$

$$\text{elevation uncertainty}_{\text{interpolated grids}} = 2 \times \sqrt{\text{variance error}}$$

where $b_i$ is the elevation, SE ($b_i$) is the standard error of the elevation, and t (1-0.025, n-p) is the 95% percentile of t-distribution with n-p degrees of freedom, n is the number of ICESat-2 measurements in one grid, p is the number of regression coefficients (7) in the text.

The slope was calculated by the method of Horn et al. (1994). Based on the law of propagation, the slope uncertainty was calculated as follows:

$$\text{slope} = \frac{\sqrt{[(e_1 + 2e_4 + e_6) - (e_3 + 2e_5 + e_8)]^2 - [(e_6 + 2e_7 + e_8) - (e_1 + 2e_2 + e_3)]^2}}{8d}$$

$$\text{slope uncertainty} = \sqrt{\sum_{i=1}^{8} (\frac{\partial_{slope}}{\partial_{e_i}} \times \sigma e_i)^2}$$

where $e_2$, $e_4$, $e_5$, $e_7$ are the elevation values adjacent to the central pixel, $e_1$, $e_3$, $e_4$, $e_6$ are the elevation values on the diagonal of the central pixel, $\sigma e_i$ is the elevation uncertainty of the corresponding pixel.

The above statement has also been added into the manuscript.

7.In general, the labels of the figures are arranged in the paper from small to large, please check the order of Figure 4 and Figure 5.
    Responses: We have checked the order of figures, and exchanged Figure 5 and Figure 6 according to the revised manuscript.

8.The structure of the conclusion and discussion section is a bit confusing, please reorganize it. Generally speaking, accuracy verification is part of the results.

Responses: We have moved the accuracy verification to the results, and reorganized the text as you suggest.

---

## Author Response (AR2)

We are satisfied with the revisions. I strongly agree with the reviewer's assessment of this article that it is very meaningful to derive a new Greenland DEM from ICESAT-2. I am pleased to inform that the paper is now can be accepted for publication with some minor revisions as follows:

We thank you for the helpful feedback, these suggestions have significantly improved the text and figures, we are appreciative of your help and time.

1. P10, in the part of "Date availability": "National Tibetan Plateau Data Center, …" should be changed to "National Tibetan Plateau/Third Pole Environment Data Center, …". A detailed description of the data center can be found in this article: Li X, Che T, Li XW, Wang L, Duan AM, Shangguan DH, Pan XD, Fang M, Bao Q. CASEarth Poles: Big data for the Three Poles. Bulletin of the American Meteorological Society, 2020, 101(9): E1475-E1491, 10.1175/BAMS-D-19-0280.1.

Responses: Accept and revised.

2. In the Part "5. Comparison with other available DEMs": The results show that the accuracy of ICESat-2 DEM is comparable to the 500 m ArcticDEM (Table 5 and 6), however the comparison is done at different time periods and the grid number is different. Since both data are available for the year 2018, can you do a comparison of these two data using the same grid number for the same time period? I think the results of this comparison can better illustrate the advantages of the new data.

Responses: Accept and revised. We did a comparison of 500 m ArcticDEM and ICESat-2 DEM by using IceBridge data of year 2018. The same conclusion can be drawn that ICESat-2 DEM is comparable to the 500 m ArcticDEM. The results are as follows.

Table 7: Elevation differences of the ICESat-2 DEM and 500m ArcticDEM with respect to IceBridge data in the entire Greenland and stable regions which have little elevation change rate.

| Region | DEM (grid numbers) | MED (m) | MD (m) | MAD (m) | STD (m) | RMSE (m) | R |
|---|---|---|---|---|---|---|---|
| Entire Greenland | ICESat-2 DEM (90141) | -0.24 | -0.59 | 3.21 | 12.22 | 12.24 | 0.9999 |
| | 500m ArcticDEM (90141) | 0.49 | 1.52 | 2.07 | 8.11 | 8.25 | 0.9999 |
| Stable regions | ICESat-2 DEM (22937) | -0.11 | -0.20 | 1.31 | 6.15 | 6.16 | 0.9999 |
| | 500m ArcticDEM (22937) | -0.19 | -0.05 | 0.86 | 2.52 | 2.52 | 0.9999 |

---

## Author Response (AR3)

This paper proposed a new 500m DEM over Greenland using ICESat-2 data from November 2018 to November 2019, following the same approach as presented by Slater et.al. (2018). The new DEM is validated with OIB data and compared to existing Arctic DEMs. Results showed the ICESat-2 DEM has significant improvements in accuracy compared with other altimeter-derived DEMs and is also comparable to DEMs derived from stereo-photogrammetry and interferometry. There are three minor revisions as follows:

We thank you for the helpful feedback, these suggestions have significantly improved the text and figures, we are appreciative of your help and time.

1、 There are some doubts about the meaning of "grid numbers" in the tables of the article. It's not clear what these numbers mean.

Responses:
"Grid numbers" means the number of compared grid cells, i.e., the number of grids covered by IceBridge data. We have changed the table column name in the revised version.

2、 Table 4.2 would be nice to supplement the effect of a 5 km grid, which can be contrasted with Interpolated grids.
Responses:
We compared the grids derived from the 5 km grids, and it performed worse than the 2 km DEM but still showed a better performance than the interpolated grids.

Table R1: Elevation differences between the ICESat-2 DEM and IceBridge data under different DEM resolutions.

| Resolution (grid numbers) | MED (m) | MD (m) | MAD (m) | STD (m) | RMSE (m) | R |
|---|---|---|---|---|---|---|
| 500m (11186) | -0.09 | -0.16 | 0.60 | 2.55 | 2.55 | 0.9999 |
| 1km (6903) | -0.01 | -0.04 | 0.71 | 2.81 | 2.81 | 0.9999 |
| 2km (8453) | -1.37 | -1.78 | 2.52 | 6.34 | 6.59 | 0.9998 |
| 5km (245) | -0.72 | -1.39 | 6.89 | 14.18 | 14.24 | 0.9998 |

3、 Table 7 shows the comparison of ICESat-2 DEM with the 500 m ArcticDEM. The MED and MD of the ICEsat-2 DEM are smaller than those of the 500 m ArcticDEM, but the MAD, STD, and RMSE are all larger than those of the 500 m ArcticDEM. Can we conclude that the new results are better than the old one?

Responses:
ICESat-2 DEM showed significant improvements in accuracy compared with other altimeter-derived DEMs and is also comparable to DEMs derived from stereo-photogrammetry and interferometry.
The MAD, STD, and RMSE are all larger than those of the 500 m ArcticDEM, and it is reasonable that stereo-photogrammetry can generate more consistent elevation

estimations at the regional scale than altimetry. Nevertheless, the ICESat-2 DEM is comparable to DEM and 500 m ArcticDEM when slopes are less than 1°, which occupies approximately 70% of Greenland. In addition, compared to the 500 m ArcticDEM, the ICESat-2 DEM can provide an elevation reference with a definite time stamp, which is essential for further ice dynamics and mass change estimation.